

# Adaptive memory reservation strategy for heavy workloads in the Spark environment

Bohan Li[1], Xin He[1], Junyang Yu[1], Guanghui Wang[1], Yixin Song[2], Shunjie Pan[1] and Hangyu Gu[1]

[1] School of Software, Henan University, Kaifeng, Henan Province, China
[2] School of Computer Science and Engineering, Southeast University, Nanjing, Jiangsu Province, China

Corresponding author
Xin He, hxsyjkf@foxmail.com

## ABSTRACT

The rise of the Internet of Things (IoT) and Industry 2.0 has spurred a growing need for extensive data computing, and Spark emerged as a promising Big Data platform, attributed to its distributed in-memory computing capabilities. However, practical heavy workloads often lead to memory bottleneck issues in the Spark platform. This results in resilient distributed datasets (RDD) eviction and, in extreme cases, violent memory contentions, causing a significant degradation in Spark computational efficiency. To tackle this issue, we propose an adaptive memory reservation (AMR) strategy in this article, specifically designed for heavy workloads in the Spark environment. Specifically, we model optimal task parallelism by minimizing the disparity between the number of tasks completed without blocking and the number completed in regular rounds. Optimal memory for task parallelism is determined to establish an efficient execution memory space for computational parallelism. Subsequently, through adaptive execution memory reservation and dynamic adjustments, such as compression or expansion based on task progress, the strategy ensures dynamic task parallelism in the Spark parallel computing process. Considering the cost of RDD cache location and real-time memory space usage, we select suitable storage locations for different RDD types to alleviate execution memory pressure. Finally, we conduct extensive laboratory experiments to validate the effectiveness of AMR. Results indicate that, compared to existing memory management solutions, AMR reduces the execution time by approximately 46.8%.

## INTRODUCTION

As the volume of data on the Internet continues to rise, coupled with the demand for real-time big data processing, Spark is experiencing exponential growth in the amount of processed data. Effective management and suitable data analysis tools are imperative for harnessing the potential benefits of big data in various organizational and industrial domains (*Adinew, Shijie & Liao, 2020*; *He et al., 2022*). Spark, a widely adopted open-source distributed big data processing framework. Leveraging in-memory computing, it enhances real-time data processing in the big data environment, ensuring both high

fault tolerance and scalability (*Cheng et al., 2021*; *Feng & Wang, 2020*; *Yoon & Kim, 2020*). It is characterized by speed, user-friendly features, versatility, and compatibility (*Yang, Wang & Fu, 2021*; *Hoseinyfarahabady et al., 2020*). Spark employs a directed acyclic graph (DAG) for job scheduling, breaking down diverse jobs into numerous subtasks based on dependency relationships. This approach minimizes the transfer process between different jobs. The DAG is constructed from the relationships established by RDD operators and resilient distributed datasets (RDD) (*Sidhanta, Golab & Mukhopadhyay, 2019*; *Tang et al., 2018*). RDD, a pivotal abstraction in Spark, represents a collection of objects distributed across nodes in the Spark cluster. By storing these RDDs in memory, the need for frequent data read and write operations from external storage media is eliminated.

The growing volume and complexity of big data have led to an increase in the size and number of RDDs, resulting in significant memory pressures in the Spark environment. Efficient memory management has become a crucial research challenge for Spark, aiming to utilize memory resources effectively (*Shi et al., 2020*; *Tang et al., 2020*). Initially, Spark employs a static memory management mechanism, fixing the size of storage memory, execution memory, and other memory throughout the Spark application's duration (*Apache, 2024*). The static memory mechanism lacks a method for dynamic space allocation, leading to unbalanced space allocation in the presence of varying memory occupancy (*Wang, 2021*). Furthermore, unified memory management has been introduced since Spark version 1.6 (*Zaharia et al., 2016*; *Apache, 2019*). The unified memory management mechanism in Spark has enhanced the utilization of both in-heap and out-of-heap memory resources to a certain extent, reducing the challenges for developers in managing Spark memory (*Wang, 2021*).

However, Spark's memory management continues to encounter challenges in computational efficiency for heavy workloads scenarios. Task blocking and garbage collection in memory management significantly affect computational efficiency in the Spark environment. Figure 1 presents experimental tests demonstrating the computational efficiency of Spark for default persistence strategies: only_memory, memory_and_disk, and only_disk (*Zhang et al., 2018*; *Zhao et al., 2019*; *Wang et al., 2018*). The experimental tests use the PageRank workload. Computational efficiency is illustrated based on the choice of RDD persistence strategy for various workloads. Figures 1A and 1B depict the impact of RDD persistence strategy choice on task completion time and garbage collection percentage. It results in additional time wastage due to garbage collection and re-computation of RDDs that are evicted from the cache. Hence, exploring advanced memory management strategies for heavy workloads scenarios is crucial for enhancing the computational efficiency of Spark.

Previous studies have explored memory management to enhance Spark's computational efficiency. Both DMATS and DMAOM strategies are advanced optimization approaches for enhancing Spark's computational efficiency in scenarios with large data volumes deployed on physical multi-node servers. *Tang et al. (2020)* proposed a dynamic Spark memory-aware task scheduler (DMATS). Dynamically adjust task scheduling based on memory usage, changing task parallelism in memory, and improve task execution efficiency. *Wang et al. (2019b)* introduced the on-demand memory allocation strategy DMAOM. It allocates

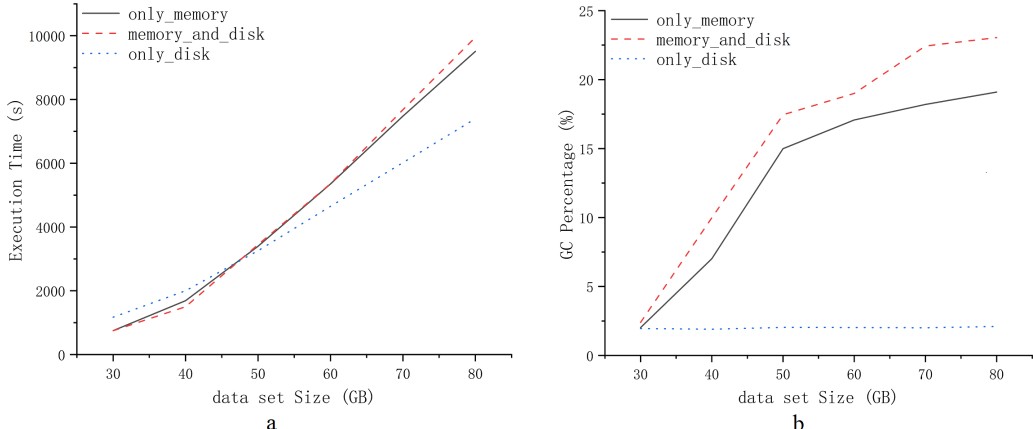

**Figure 1** Experiments comparing three default persistence strategies for Spark.

memory proportionally from the resource pool based on the task's requested memory size. Nevertheless, it does not quantify the minimum memory resources required for task operations. Therefore, further research on memory management is necessary to enhance Spark's computational efficiency for heavy workloads scenarios. In this article, we present an adaptive memory reservation (AMR) strategy aimed at enhancing computational efficiency for heavy workloads in the Spark environment. The core idea is to dynamically adjust the execution memory space by compressing or expanding it, ensuring a designated amount of storage memory space is reserved while maximizing task parallelism. Simultaneously, the disk is utilized as supplementary storage for RDDs, ensuring a high persistence hit rate. Experimental validation has demonstrated that utilizing the disk can mitigate contention between execution memory and storage memory, reduce memory pressure, optimize garbage collection percentages, and enhance computational efficiency (*Wang et al., 2019a*; *Li et al., 2018*; *Bae et al., 2017*; *Singer et al., 2011*; *Xu et al., 2016*; *Song et al., 2023*). The AMR strategy comprises three main steps. Firstly, it assesses whether the memory space meets the conditions for task parallelism optimal memory (TPOM) execution. If a task is identified as blocked, execution memory is reallocated with a priority on maintaining task parallelism. Secondly, the memory adaptive adjustment algorithm is activated, and excess memory is reclaimed when tasks in memory are running normally. Thirdly, the RDD is controlled to select a low-cost computation location based on the allocated memory space, ensuring sufficient room to prioritize the space reserved for execution memory. The primary contributions of this paper can be summarized as follows:

- We formulate an optimal task parallelism problem aimed at minimizing the disparity between the number of tasks completed without blocking and those completed in regular rounds. The proposed optimal task parallelism strategy aims to allocate memory space based on the characteristics of Spark tasks, ensuring computational parallelism and enhancing resource utilization efficiency.
- We dynamically ensure task parallelism by applying adaptive execution memory reservation, choosing to compress or expand the allocated execution memory space

based on task execution dynamics. We recommend selecting suitable storage locations for various RDD types to alleviate execution memory pressure, considering the cost of RDD cache location and real-time memory space usage.

The following sections of this article are structured as follows: Section 'Related works' reviews related work, Section 'Spark model and problem descriptions' introduces models and optimization goals, Section 'Adaptive memory reservation strategy' outlines the data scheduling strategy for adaptive memory, Section 'Experimental evaluation' details the experimental results. Finally, Section 'Conclusion' concludes the paper.

## Related works

With the increasing impact of the Spark distributed computing framework, there has been a gradual emergence of research on memory management strategies in parallel computing frameworks (*Tang et al., 2020*; *Swain, Paikaray & Swain, 2011*; *Duan et al., 2016*; *Meng et al., 2017*; *Yun & Yuchen, 2020*; *Chen et al., 2016*; *Wang et al., 2019b*; *Song YX, 2022*). Various strategies aim to enhance operational efficiency through memory optimization: *Swain, Paikaray & Swain (2011)* introduced the concept of weighted memory replacement and proposed the AWRP algorithm based on weights, but did not specifically optimize it for Spark. Meanwhile, *Duan et al. (2016)* suggested a diversified weighted cache replacement method that selects partitions by comprehensively considering computational cost, usage frequency, and size.

*Meng et al. (2017)* considered additional RDD characteristics in weighted replacement algorithms, achieving positive outcomes when sufficient memory and network bandwidth were available. However, partition selection involves network communication, and the algorithm places high demands on memory and network performance, leading to relatively lower when dealing with large data volumes and insufficient cluster performance.

The ERAC algorithm proposed by Wei et al. (*Yun & Yuchen, 2020*) prevents memory leaks by optimizing the weighted model and replacement strategy, calculating the size of each RDD in advance, and defining a Boolean value to exclude incomplete RDD partitions. However, the drawback is that the algorithm itself occupies too much memory. *Song, (2022)* proposed a new concept of utility entropy, which can better measure the degree of match between checkpoint caching and memory. By finding the best match and selecting the appropriate time to perform checkpoint cleaning, it improves both memory utilization and fault tolerance efficiency. *Li et al. (2023)* based on the weight relationship established by the sample lineage graph of RDD, further improve computing efficiency by increasing the complexity of the weights. However, it inevitably runs into a severe memory shortage, which leads to garbage collection. They also focused on optimizing the checkpoint caching and cleaning process in Spark.

Some experts recommend utilizing I/O transfers to enhance CPU utilization (*Song et al., 2022a*; *Tang et al., 2021*). Nevertheless, optimization is constrained by network bandwidth and cannot exceed. *Song et al. (2023)* introduced MCM, an optimization for persistent acceleration at the application level.

*Chen et al. (2016)* conducted an analysis, comparing the similarity of extensive historical task feature data. They predicted the most suitable parameters for the Spark job before

submission, determining the optimal values. However, its limitations are apparent, as it necessitates optimization for the same type or even the same Spark task that has been previously executed. SP-LRP proposed by *Huang (2020)* predicts the size of reduce task partitions, this approach uses the skew detection algorithm to identify skewed partitions and adjusts task resource allocation based on the fine-grained resource allocation algorithm. However, these predictions demand additional computing resources.

Other optimization strategies for Spark include those based on parameter tuning. SMBSP proposed by *Rahman, Hossen & Venkataseshaiah (2018)*) is a machine learning-based optimization strategy for Spark with remarkable performance improvements. Its strategy fundamentally differs from cache strategies based on the Spark framework, placing it in a different optimization category. Moreover, SMBSP exhibits consistent performance improvements across different data sizes, suggesting its compatibility with other framework optimization strategies. MLAT, proposed by *Nikitopoulou et al. (2021)*, is an automatic parameter optimization driver. It does not compete in the same area as cache strategies based on the Spark framework, and combining the two is likely to yield greater improvements. We will continue to monitor research in this direction.

*Zhu et al. (2017)* proposed a memory-constraint strategy to reduce frequent cache eviction and substantial garbage collection time. It alleviates memory pressure to a certain extent, but the execution performance for larger data volumes has not been verified, and the paper does not demonstrate the memory usage. *Xu et al. (2016)* abstracted the task execution memory and RDD cache space as a memory contention issue, proposing dynamic adjustments to their ratio to improve execution efficiency. This method is only applicable to memory contention issues with relatively small data volumes, as indicated by the memory usage in the paper; the experimental data used do not lead to severe memory overflow. DSMM (*Chae & Chung, 2019*), through testing various strategies for different data sizes, derives an algorithm that selects different caching strategies based on data size, adapting to some extent to computations for varying data volumes. However, this method is relatively rigid; for example, it cannot fully leverage memory space for heavy workloads. It is not a universally applicable approach.

Currently, optimization approaches for Spark memory are relatively limited, lacking broad optimization for various data volumes. They always focus on large data volumes without fully leveraging the remaining memory space. The AMR strategy consistently ensures the optimal utilization of the reserved space, regardless of the data volume.

## SPARK MODEL AND PROBLEM DESCRIPTIONS

Insufficient memory can lead to various issues, including memory contention, garbage collection, and task blocking. Section 'Spark task flow and memory' displays the Glossary of symbols and terms in Table 1, illustrating the relationship between Spark task flow and memory. In Section 'Optimal task parallelism', we model the optimization goals of Spark task flow and derive the optimization objective function. Finally, in Section 'Adaptive memory reservation', we present a data dispatching model based on adaptive reserved memory, along with a formal description of the adaptive reservation strategy.

**Table 1  Glossary of symbols and terms.**

| Variable name | Description |
|---|---|
| $R_T$ | Target RDD |
| $fR_i$ | Parent RDD of the i-th RDD |
| $f_f R_i$ | Parent RDD of the i-th parent RDD |
| $R_{size}$ | Current RDD size |
| $fR_{size}$ | Parent RDD size of RDD |
| $\rho_c()$ | Reading cost |
| $\chi_c()$ | Calculation cost |
| $\Phi_n$ | Active task number |
| $persistState$ | Current persistence state |
| $V_{memo}$ | Memory transfer speed |
| $V_{disk}$ | Hard disk transfer speed |
| $T$ | Time cost for task calculation |
| $T'$ | Time cost for target RDD calculation |
| $T_G$ | Time cost of garbage collection |
| $M_U$ | Size of memory usage |
| $M_S$ | Size of storage memory usage |
| $M_P$ | The amount of memory space that needs to be satisfied for the task to run |
| $M_E$ | Size of execution memory space |
| $M_M$ | Maximum available memory space |
| $N_D$ | Number of tasks actually completed during normal execution rounds |
| $N_E$ | Number of executors |
| $N_C$ | Number of executor cores |
| $N$ | Number of tasks completed without blocking in normal execution rounds |
| $n$ | Number of execution rounds |
| $\tau$ | Running tasks in the current execution memory |
| $\Lambda$ | spark.storage.storageFraction, which controls the size of Storage memory |
| $\Psi$ | Probability of RDD being replaced out in memory |
| $\kappa$ | Learning rate |
| $\sigma$ | Threshold to measure memory usage |

## Spark task flow and memory

The Spark task execution process primarily involves four steps: First, building the directed acyclic graph (DAG) by calling methods on RDDs. In Spark, the entire execution process logically forms a DAG, and after the Action operator is triggered, all the accumulated operators are formed into a DAG based on the lineage of the RDD. Second, the DAGScheduler divides the DAG into stages based on Shuffle and sends the generated tasks in each stage to the TaskScheduler in the form of a TaskSet. Subsequently, the TaskScheduler schedules tasks. Finally, the Executor of each Worker node receives tasks and puts them into a thread pool for execution. Figure 2 illustrates that RDDs are transformed

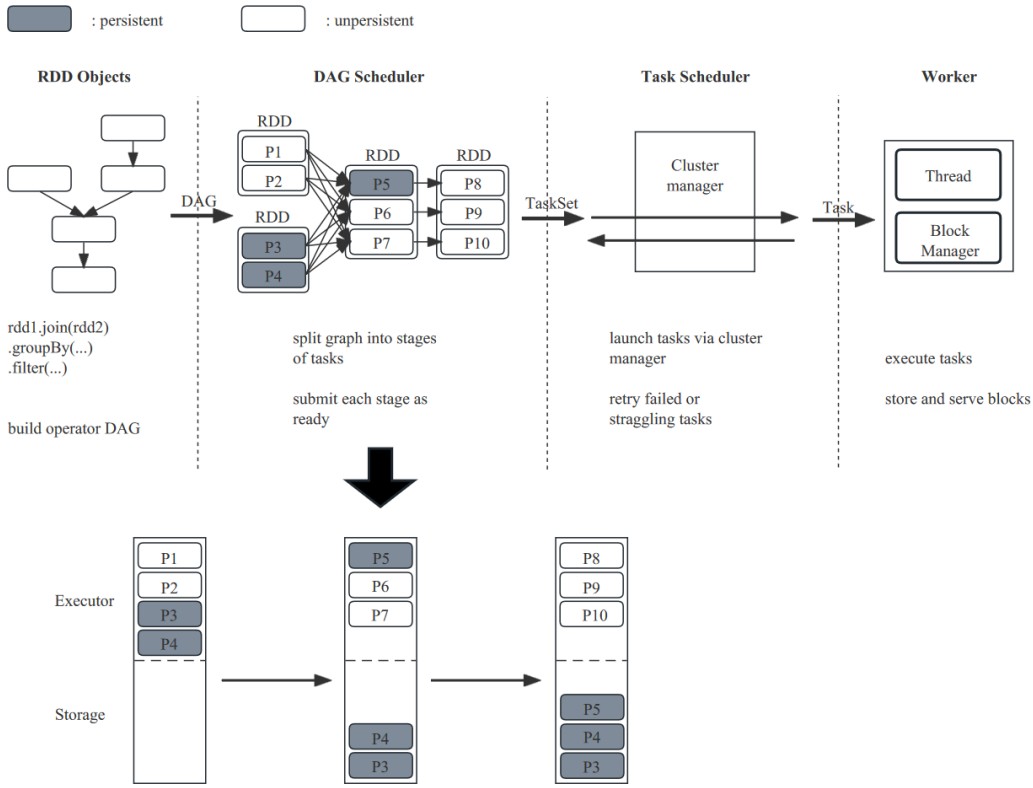

**Figure 2** The execution process of Spark tasks (RDDs are stored in local memory as RDD partitions, *i.e.,* P1, P2).

based on their dependencies, and the resulting RDD partitions are stored in memory after the transformation. If the submitted data is excessively large, tasks may be blocked due to insufficient memory or RDD eviction. Thus, our optimize based on the characteristics of the task runtime state and the DAG graph to ensure tasks can be executed while maximizing the RDD persistence rate.

To address the problem of uneven memory space allocation, Spark introduced Unified Memory Management after version 1.6. Leveraging the Unified Memory Management mechanism has enhanced the utilization of both heap and off-heap memory resources, reducing the complexity for developers in maintaining Spark memory. However, developers still need to allocate memory space reasonably. For example, if the storage memory space is too large, Spark will frequently perform full garbage collection, thereby reducing performance during task execution. Cached RDD data typically resides in memory for an extended period. To fully leverage Spark's performance, further optimization of the memory allocation strategy is necessary.

## Optimal task parallelism

Computation in memory cannot proceed as intended when the memory is flooded with data. Insufficient memory can lead to severe memory contention. Memory contention can result in full garbage collection, leading to task suspension. Severe memory contention can

lead to task blocking. In this section, we formally describe the task blocking problem and present the objective function.

Tasks running in execution memory share execution memory, in Spark, the HashMap structure holds the mapping of tasks to memory consumption. The range of execution memory size that each task can occupy is $\frac{1}{2\Phi_n} \sim \frac{1}{\Phi_n}$, where $\Phi_n$ is the number of tasks currently running in the execution memory. When each task starts, it needs to request the memory manager to apply for a minimum execution memory of $M_P = \frac{M_E}{2 \times \Phi_n}$ size, if the requirements cannot be met, the task is blocked until other tasks release enough execution memory, and the task can be reawakened.

When there are tasks blocked, the number of tasks executed in this round is reduced compared with the number of tasks executed in the normal round, and additional rounds are required to execute these unfinished tasks. In the actual operation process, it is unavoidable that tasks are blocked due to insufficient execution memory. Let $\{\tau_1, \tau_2, \tau_3, \ldots, \tau_{\Phi_n}\}$ be a set of active task, the number of tasks actually completed in a normal round is shown in Eq. (1).

$$N_D = n \times \Phi_n \tag{1}$$

where $N_D$ represents the number of tasks actually completed in a normal round. $n$ represents the number of execution rounds and $\Phi_n$ is the number of tasks currently running in the execution memory. We set $N$ to be the number of tasks that should have been executed if the task did not block when the memory is sufficient. The original number of tasks that should be completed without blocking is shown in Eq. (2).

$$N = n \times (N_E \times N_C) \tag{2}$$

where $N$ represents the number of tasks completed without blocking, $N_E$ represents the number of executors, and $N_C$ represents the number of cpu cores in a single executor. For the default configuration of Spark, when the execution memory is crowded and cannot meet the minimum memory requirements of the current execution task, the task will be blocked and the corresponding number of tasks cannot be completed in the normal round, so, $N_D$ will be less than $N$. The additional time to complete these blocked tasks is a waste of time caused by insufficient memory. Based on the above description, we give a formal description of the task blocking problem caused by insufficient memory, and establish the optimization objective function of the problem as shown in Eq. (3).

$$min(N - N_D), (N > N_D) \tag{3}$$

Where $N$ represents the number of tasks completed without blocking and $N_D$ represents the number of tasks actually completed in normal rounds, and the optimal task parallelism is achieved when the value of the difference between the two is minimum. $min$ represents the minimum value. We minimize the absolute value of the difference between the number of tasks completed in the non-blocking case and the number of tasks actually completed in a normal round as the optimal solution.

## Adaptive memory reservation

To optimize the objective function (Eq. (3)), we introduce an adaptive memory reservation data dispatching strategy. This strategy ensures task execution parallelism and actively reduces the overall task execution time by overflowing RDDs and reserving memory space. In this section, we provide a formal description of this strategy model.

RDD serves as Spark's foundational data abstraction, comprising a collection of partitioned records. It can generate new RDDs on stable physical storage or apply transformations to existing RDDs. The transformed RDDs retain dependencies on the original RDDs, forming a lineage. Records the lineage series that created the RDD, enabling the recovery of lost partitions. The RDD's lineage records metadata information and transformation behavior. If a partition data of this RDD is lost, it can recompute and recover the lost data partition using this information. The RDD generation process begins by reading its parent RDD first. Each RDD is created along the directed acyclic graph path through a series of conversion operations. We refer to the parent RDD of an RDD as $fR$. Due to wide and narrow dependencies between RDDs, an RDD may have multiple $fR$ simultaneously, and the persistence status of different $fR$s may vary. The calculation cost of an RDD is represented by Eq. (4).

$$\chi_c(R_T) = \sum_{k=1}^{u} \rho_c(fR_k) + \lambda \qquad (4)$$

where $\chi_c(R_T)$ denotes the calculation cost of the target RDD, $fR$ denotes the data set of a parent RDD, the $\sum_{k=1}^{u} \rho_c(fR_k)$ denotes the reading cost of all $fR$s, and the $\lambda$ denotes the calculation cost of all $fR$s transformation to the target RDD. $u$ denotes the number of parent nodes of the target RDD, and $k$ denotes each parent node.

The actual reading cost is not only the cost of transmission. Memory contention caused by insufficient memory can lead to frequent garbage collection, and the impact of garbage collection time on the overall task execution time cannot be ignored. Moreover, RDDs that are evicted from memory need to be re-read from their parent RDDs and transformed. We set there have $w$ RDDs in $N$ tasks. Based on the objective function (Eq. (3)) and the calculation cost of the target RDD (Eq. (4)), we can formalize the total task cost T as shown in Eq. (5).

$$T = \sum_{s=1}^{w} \chi_c(R_{T_s}) \qquad (5)$$

where $\chi_c(R_T)$ is the calculation cost of the target RDD, $s$ denotes each target RDD.

When a task reads a partition at the beginning of startup, it will first determine whether the partition has been persisted. If not, it needs to check the checkpoint or recalculate according to the lineage. if multiple actions are to be performed on an RDD, this RDD can be persisted in memory or on disk using persist or cache methods in the first action. If $fR$ is persisted, $\rho_c(fR)$ is equivalent to data transfer cost. Let $f_f R$ be the parent RDD of $fR$, the $\chi_c(fR) = \sum_{k'=1}^{u'} \rho_c(f_f R_{k'}) + \lambda'$. If $fR$ is not persisted, then $\rho_c(fR)$ is equivalent to the calculation cost of $fR$. Each RDD has a variety of dependencies with its parent RDD.

For comprehensive consideration, we maintain the possibility of different RDDs based on persist state. There is still a parent RDD that needs to be recalculated, so the $\rho_c(fR)$ is shown in Eq. (6).

$$\rho_c(fR) = \begin{cases} \chi_c(fR), & persistState = false \\ \dfrac{fR_{size}}{v_{memo}}, & persistState = memory \\ \dfrac{fR_{size}}{v_{disk}}, & persistState = disk \end{cases} \tag{6}$$

Where $persistState$ denotes the persistence state of $fR$, $fR_{size}$ denotes the size of $fR$, $v_{memo}$ denotes the transfer speed of memory and $v_{disk}$ denotes the transfer speed of disk. When $persistState = false$, it means the RDD unpersist, $\rho_c(fR) = \chi_c(fR)$. When $persistState = memory$ the RDD persisted in memory, $\rho_c(fR) = \frac{fR_{size}}{v_{memo}}$. When $persistState = disk$ the RDD persisted in disk, $\rho_c(fR) = \frac{fR_{size}}{v_{disk}}$.

## RDD location selection

We have designed a cache location selection model to proactively and dynamically select the cache location of RDDs (Eq. (10)). From the experiments mentioned above (Fig. 1), we can see that persisting RDDs continuously in memory or on disk can eventually lead to unilateral performance degradation. We can further specify $\chi_c(R_T)$ as shown in Eq. (7).

$$\chi_c(R_T) = \sum_{r=1}^{u-P-Q} \chi_c(fR_r) + \sum_{p=1}^{P} \frac{fR_{p_{size}}}{v_{memo}} + T_G + \sum_{q=1}^{Q} \frac{fR_{q_{size}}}{v_{disk}} + \lambda \tag{7}$$

where $u$ represents the number of $fR$s, $P$ represents the number of $fR$s which persisted in memory, $Q$ represents the number of $fR$s which persisted in disk and the $T_G$ represents the cost of garbage collection caused by $fR$ persisted in memory. Therefore, we can make full use of the space in memory and on disk. When the speed of memory persistence is faster, we persist RDDs in memory, and vice versa, we persist them on disk. Combining the cost of $\rho_c(fR)$ (Eq. (6)), we can expand the extra task cost formula (Eq. (5)) as a sub-problem shown in Eq. (8).

$$T' = min(\chi_c(R_T), \frac{R_{T\,size}}{v_{memo}} + T_G, \frac{R_{T\,size}}{v_{disk}}). \tag{8}$$

The sub-problem is solved when the current RDD selected storage location. The $T'$ represents the cost of target RDD when choose different persistState, where $\chi_c(R_T)$ represents the cost when target RDD unperist (Eq. (2)), $\frac{R_{T\,size}}{v_{disk}}$ represents the time cost when target RDD persisted in disk (Eq. (2)), $\frac{R_{T\,size}}{v_{memo}} + T_G$ represents the time cost when target RDD persisted in memory (Eq. (2)), and $T_G$ is the garbage collection cost caused by RDD which persisted in memory.

## ADAPTIVE MEMORY RESERVATION STRATEGY

In summary, our algorithm accounts for the effects of execution memory on task blocking, the influence of garbage collection due to cache contention, and the repercussions of

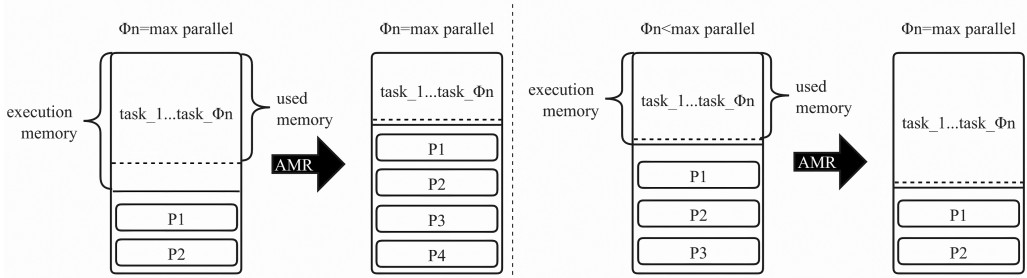

**Figure 3  Settings for execution memory in different situations.**

different cache media transfer rates on the cost of RDD reads when altering the cache strategy. We design the AMR algorithm to replace Spark's default dynamic caching mechanism, aiming to minimize cache contention while ensuring task parallelism (Fig. 3). Simultaneously, we alleviate the burden on storage memory by dynamically adjusting the cache location strategy to store less crucial RDDs on the disk, thereby reducing garbage collection costs and the likelihood of more vital RDDs being evicted from the cache (Fig. 4).

In Fig. 3, we determine whether tasks in all execution memories have achieved maximum parallelism. If the tasks reach this threshold, and there is available space in the execution memory, we allocate memory to satisfy the minimum requirement for all tasks. At this point, the storage memory space increases, allowing for the persistence of more RDD partitions. If the running tasks do not reach maximum parallelism, it signals inadequate space in the execution memory. We calculate the minimum memory required for a single task ($M_P$) and the execution memory size needed for the task at maximum parallelism. This calculation is based on the number of running tasks ($\Phi_n$) and the current execution memory size ($N_E \times N_C$). The RDD partition is populated to reserve storage memory space of the corresponding size, expanding the execution memory to meet the size required for the task at maximum parallelism. As expressed in Eq. (9).

$$M_E = \begin{cases} \dfrac{(M_U - M_S)(\Phi_n + 1)}{2\Phi_n}, & M_U - M_S < M_E, \Phi_n = N_E \times N_C \\ \dfrac{(M_U - M_S)(N_E \times N_C + 1)}{\Phi_n}, & \Phi_n < N_E \times N_C \end{cases} \tag{9}$$

The fundamental concept of LCLS is as follows. As shown in Fig. 3, the disk is used as storage space for various types of RDDs to prevent redundant RDD computations. When changing the cache strategy, factors such as the effect of execution memory on task blocking, the impact of garbage collection due to cache contention, and the influence of different cache media transfer rates on the cost of RDD reads are taken into account. The AMR replaces Spark's default dynamic caching mechanism to minimize cache contention and ensure task parallelism. Simultaneously, the burden on storage memory is reduced by dynamically adjusting the cache location strategy. Persist less critical RDDs on the disk to decrease garbage collection time and the likelihood of more crucial RDDs being evicted from the cache.

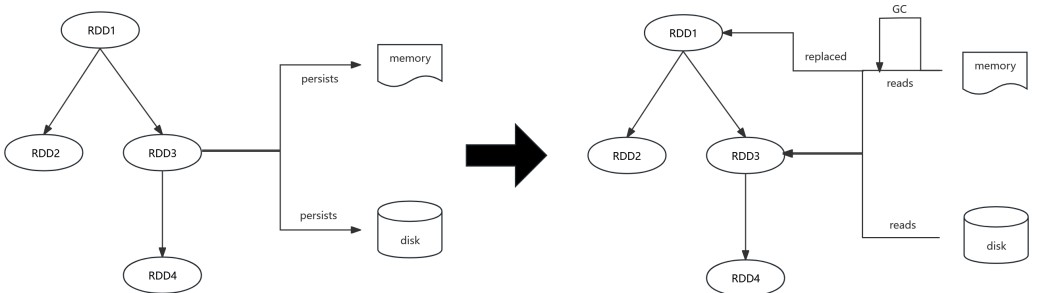

**Figure 4** Spark persists and reads RDDs in different locations.

## Adaptive memory reservation

This section provides details about the Adaptive Memory Reservation (AMR) algorithm. Algorithm 1 presents the AMR algorithm, outlining its collaboration with other algorithms and the calling relationships between the Task Parallelism Optimal Memory (TPOM) algorithm, the Adaptive Memory Tuning (AMT) algorithm, and the Low-Cost Location Selection (LCLS) algorithm. Algorithm 2 provides the implementation details of TPOM, which is utilized to allocate an appropriate amount of execution memory space to ensure sufficient space for tasks. Algorithm 3 presents the AMT algorithm, responsible for managing the reserved memory space and dynamically adjusting memory usage based on real-time assessments of memory consumption. Algorithm 4 describes the LCLS algorithm, which allocates RDD storage locations and selects the storage location for RDDs by comparing the costs associated with different storage locations.

---

**Algorithm 1:** AMR Algorithm

**Input:** $\Phi_n, N_E, N_C, M_U, M_S, M_M$.

1 **while** *State of the process is running* **do**
2     **if** $\Phi_n < (N_E \times N_C)$ *or* $\frac{(\Phi_n+1)M_U - M_S}{\Phi_n} > M_M$ **then**
3         do TPOM
4     **else** $\frac{(\Phi_n+2)M_U - 2M_S}{\Phi_n} < M_M$
5         do AMT
6     **end**
7     do LCLS
8 **end**

---

Algorithm 1 gives the architecture of the adaptive memory reservation algorithm, which internally consists of TPOM, AMT and LCLS. When a task is found to be blocking, the first condition is satisfied and the free memory area is satisfied, TPOM is executed. Else, if there is no blocking and sufficient space left, AMT is executed, and finally, LCLS is executed to select the storage location for the RDD to reduce the memory pressure. The specific algorithm is discussed below.

## Task parallelism optimal memory

According to the properties of Spark's Executor memory (Eq. (2)), we can know the maximum number of tasks running, and by comparing the number of currently running tasks with the maximum number of tasks, we can find out if the tasks are blocking (Eq. (1)). If there are fewer tasks blocking, the value of $N - N_D$ converges to zero according to Eq. (3), and therefore the less time is wasted. The Spark port allows us to monitor the performance of Spark by obtaining real-time values of different parameters.

---

**Algorithm 2:** TPOM Algorithm

**Input:** $\Phi_n, N_E, N_C, M_M, M_S, M_E, \Lambda$.

1   **if** $\Phi_n < (N_E \times N_C)$ *and* $M_M - M_S > (N_E \times N_C) \times \frac{M_E}{2 \times \Phi_n}$ **then**

2      $M_P = \frac{M_M \times (1 - \Lambda)}{\Phi_n}$

3      $MinExeMemory = M_P \times (N_E \times N_C)$

4      $\Lambda = 1 - \frac{MinExeMemory}{M_M}$

5   **end**

---

Algorithm 2 shows the specific algorithm for TPOM. First we determine if the task is blocking and needs to be reallocated by line 1 and check if the remaining space meets the condition. Then sets $M_P$ to the minimum size of memory needed for execution (line 2), and then proceeds to calculate the size of memory needed for actual execution memory and the size of space to be allocated and executed (lines 3-4). We define Storage.storageFraction as $\Lambda$, so the product of $\Lambda$ and $M_M$ gives us the size of storage memory, and conversely the product of $(1 - \Lambda)$ and $M_M$ is the size of execution memory.

### Adaptive memory tuning

Algorithm 3 shows the core of our algorithm for adaptive memory reservation. This part mainly adjusts the memory space by judging the memory operation in real-time. Equations (6) and (8) show that the storage location of the RDD can better improve the computational efficiency by choosing the disk as the storage location when the memory space is insufficient, so we can free up more space for the task by changing the persistence location of the RDD. During operation, the memory required for a task is not constant (*Singer et al., 2011*), so space reservation for execution memory is required. First, we determine whether the current task needs space reservation or rather space reallocation (line 1), if space reservation is needed, we calculate the actual size of memory required by a single task (line 2) and then determine whether the remaining memory space meets the size of reserving a task (line 3), if it is less than the reserved space size, we perform space release(lines 5-8), if the remaining space is greater than or equal to the reserved space size (line 10), we perform a compression operation on the remaining space (lines 12-15). Our memory adaptive adjustment algorithm (AMT) ensures that the memory space is used reasonably and that not too much space is reserved for tasks. Finally, we select the

---

**Algorithm 3:** AMT Algorithm

**Input:** $\Phi_n, N_E, N_C, M_M, M_S, M_U, M_E, \Lambda$.

1 **if** $\Phi_n = (N_E \times N_C)$ **then**

2    $PerTaskMem = \frac{M_U - M_S}{\Phi_n}$

3    **if** $M_M - M_U < PerTaskMem$ **then**

4       $MemorySize = PerTaskMem + M_U - M_M$

5       **while** $cacheList.FindMinFre().size < MemorySize$ **do**

6          $cacheList.findMinFre().unpersist()$

7          $cacheList.findMinFre().deQueue()$

8          $MemorySize = MemorySize - cacheList.FindMinFre().size$

9       **end**

10    **else** $M_M - M_U \geq PerTaskMem$

11       $MemorySize = M_M - M_U$

12       **while** $MemorySize > 0$ **do**

13          $cacheList.findMinFre().persist()$

14          $cacheList.findMinFre().Queue()$

15          $MemorySize = M_M - M_U$

16       **end**

17    **end**

18 **end**

---

persistence location for RDDs that have been evicted from memory or not yet stored in memory by the LCLS algorithm, which is shown below.

## Low-cost location selection

With our persistence location selection algorithm, RDDs are persisted in memory or disk, and the relationship between $(\frac{R_{T\,size}}{v_{memo}} + T_G)$ and $(\frac{R_{T\,size}}{v_{disk}})$ is a zero-sum game problem (Eq. (8)). We need a specific parameter to determine when to choose the disk or memory. For this purpose, we set $\Psi$ as the probability of being replaced out of the persistence into memory at this time, and by calculating $\Psi$ and combining the read speed of current device memory and disk, we can know the calculation cost of persisting different locations. The probability of being replaced out of the RDD storage cache for different memory occupancy is $\Psi$. The relationship between $\Psi$ and memory utilization, garbage collection factor and RDD size can be obtained from the experimental results as follows.

$$\Psi = sigmoid(\kappa \times \frac{M_S + \sigma}{\Lambda \times M_M \times R_{size}} + \kappa') \tag{10}$$

Where $M_S$ is the storage memory usage size, $M_M$ is the maximum available heap memory space, $\Lambda$ is the spark.storage.storageFraction parameter, where $\sigma$ is the threshold to measure memory usage, $\kappa$ is a learning rate and $\kappa'$ is offset variable. When there is more free memory space, the value of $\Psi$ is closer to 0. When there is not enough memory space, the probability of RDD being replaced out of memory is infinitely close to 1.

---

**Algorithm 4:** LCLS Algorithm

**Input:** $M_M$, $M_S$, $R_{size}$, $V_{memo}$, $V_{disk}$, $\Lambda$, $R_T$.

1  **for** *RDD: RDDList* **do**

2  $\quad$ $R_{size} = RDD.count()$

3  $\quad$ $\Psi = sigmoid(\kappa \times \frac{M_S + \sigma}{\Lambda \times M_M \times R_{size}})$

4  $\quad$ **if** $\Psi \times \chi_c(R_T) + (1 - \Psi)\frac{R_{size}}{V_{memo}} \geq \frac{R_{size}}{V_{disk}}$ **then**

5  $\quad\quad$ $RDD.persist(Disk\_location)$

6  $\quad$ **else**

7  $\quad\quad$ $RDD.persist(Memory\_location)$

8  $\quad$ **end**

9  **end**

---

The pseudo-code of the low-cost location selection algorithm (LSLC) is shown in Algorithm 4. First, adds the RDD to the persistence queue (lines 1-2), and we know from Eq. (8) as well as Fig. 1B that selecting memory as the storage location for the RDD when there is not enough memory space will cause the recovery time of GC to rise linearly. To avoid the time wastage caused by the garbage collection mechanism, we select the location for storage according to equation Eqs. (5), (8), (10) to choose the location with lower storage cost for storage. Set $\Psi$ to be the probability of an RDD being evicted from memory (line 3). When the memory space is infinite, we can see that $\Psi$ is converging to zero, because all RDDs can be stored in memory at this time. When there is not enough memory space to store an RDD, the probability $\Psi$ of being evicted from memory converges to 1. Accordingly, compares the cost of storing in memory with the cost of storing on the disk (line 4), and if the cost of storing in memory is higher (line 5), the disk is chosen as the storage location. If the cost of depositing to the disk is higher (line 6), memory is selected as the storage location (line 7). This has the advantage of avoiding memory contention while significantly reducing the calculation cost of the RDD (Eq. (4)). The combination of Algorithm 3 and Algorithm 4 makes the parallelism of the tasks fully guaranteed.

## EXPERIMENTAL EVALUATION

In this section, we describe the experimental environment, benchmark load and analysis of experimental results. The experimental indicators are program execution time, garbage collection time, persistence hit rate and energy consumption.

### Experimental setup

Given that the Spark cluster is an actual cluster, generating significant data is necessary to induce memory contention. For optimal efficiency and clarity in our experiments, we carefully chose algorithms that more accurately represent Spark memory contention for comparison, as depicted in Fig. 5A. We chose five representative built-in Spark strategies: memory_and_disk_ser, memory_only_ser, memory_and_disk, only_disk and only_memory. Additionally, we compared these with other algorithms, including DMAOM, DMATS, DSMM, and MCM. Figure 5B indicates that for the WordCount

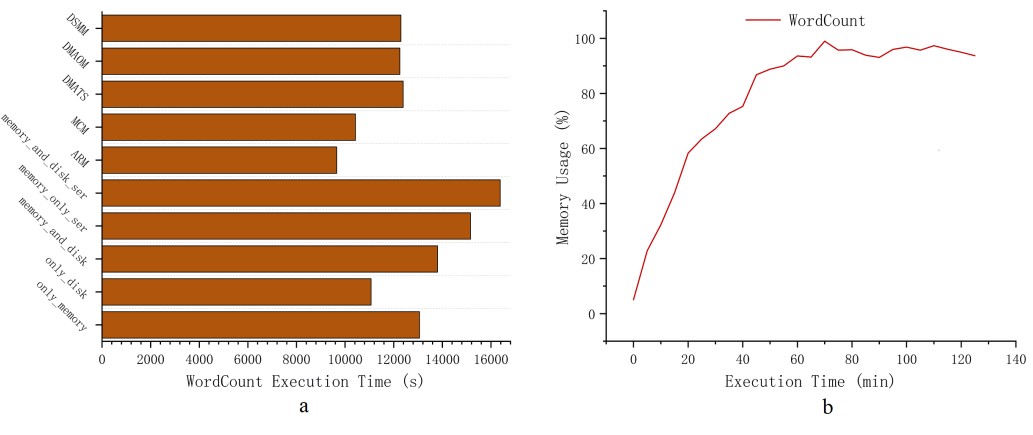

**Figure 5** Execution time.

algorithm, a data size of 900GB already results in severe memory contention. Therefore, we can conclude that this experiment is sufficient to demonstrate the computational outcomes for heavy workloads conditions. In Fig. 5A, memory_and_disk_ser and memory_only_ser fail to effectively save memory space for severe memory contention conditions, and their serialization incurs a decrease in computational efficiency due to CPU resource utilization. The MCM approach utilizes an in-memory acceleration utility algorithm, which is optimized based on the algorithm code. Unlike the independent controller of AMR, it consume computational resources for memory scheduling. Furthermore, MCM is an in-memory acceleration utility algorithm, which differs from the active spilling memory strategy aimed at enhancing computational efficiency and does not belong to the same optimization concept.DSMM innovatively takes into account the efficiency of memory compared to disk for heavy workloads. DSMM simply selects different cache locations for varying data sizes, failing to fully leverage memory acceleration and lacking consideration for RDD computation efficiency for various conditions. Therefore, it can be considered a straightforward strategy selection algorithm.

To verify the effectiveness of AMR during job execution, this section implements AMR on the Spark cluster. A variety of jobs are selected for experimentation at different data sizes to fall into memory contention to varying degrees. We define different excess data sizes according to different experimental environments and different algorithms.As shown in Fig. 6D, PageRank, K-means, and Apriori experience severe memory contention when dealing with data sizes of 80GB, 60GB, and 80GB. With the K-means algorithm, the execution time of only_memory and memory_and_disk strategies converge to the execution time of only_disk for this cluster when it is larger than 40GB. At this time, the pressure of data on memory makes Spark cluster cannot reflect the advantages of in-memory acceleration, and workload larger than this amount of data is overloaded. Similarly, with PageRank and Apriori algorithm, the execution time of only_memory and memory_and_disk strategy is converged to the execution time of only_disk for excess data when it is larger than 50GB.

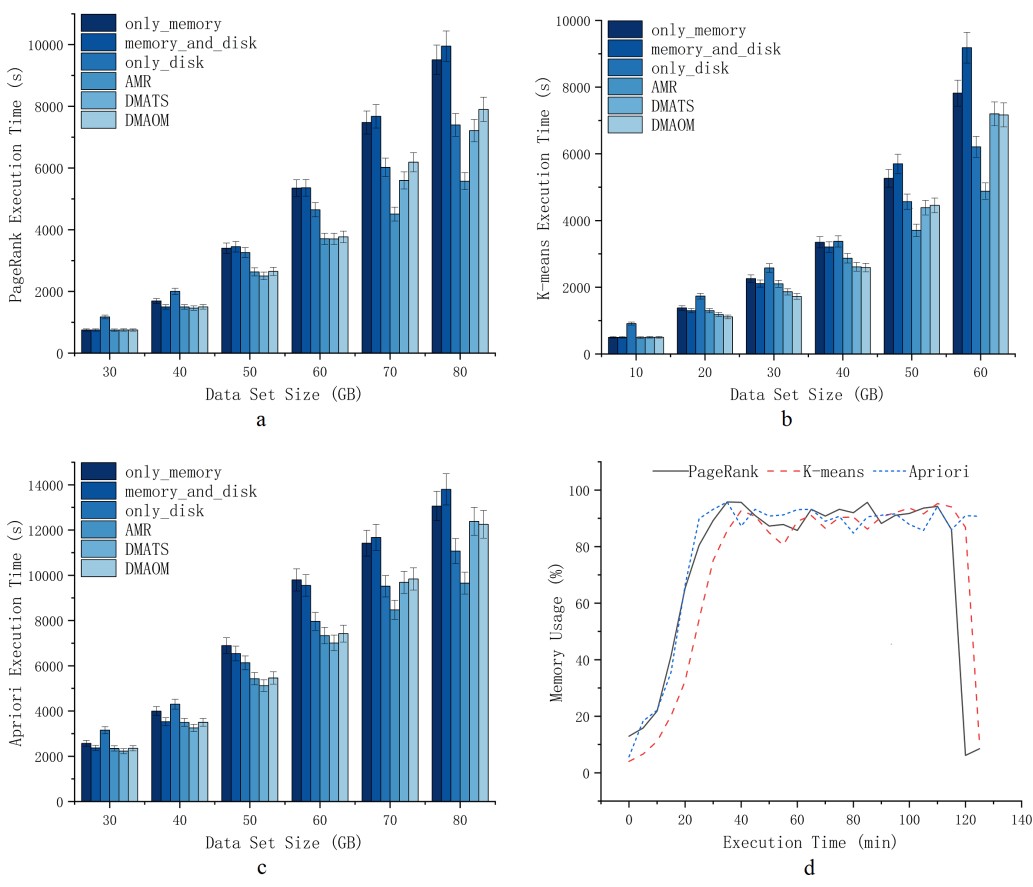

**Figure 6  Execution time.**

## Execution time

The experiment started with three worknode, and each work node start the same executor configuration. The node server configuration of the experimental environment and the Executors configuration of the node server is shown in Table 2. The experimental workloads are shown in Table 3. The experimental data are preprocessed to obtain a uniform distribution. In order to verify the execution time of different persistence locations, we set three default persistence strategies of Spark as a comparison reference in our experiments to demonstrate the effectiveness of AMR (Figs. 6A, 6B and 6C): The only_memory strategy means that all persistent RDDs are stored in the memory region. The only_disk strategy means that all persistent RDDs are stored in the disk. The memory_and_disk strategy means that persistent RDDSs are stored in memory first and overflowed to the disk when memory space is insufficient. The DMAOM strategy is to filter RDDs according to their usage, then construct an RDD structure tree and prune them, and finally choose whether to replace them according to their weights and remaining memory space. The DMATS strategy is dynamically adjust task scheduling based on memory usage, change task parallelism in memory, and improve task execution efficiency.

**Table 2  Node server configuration information.**

| Parameter | Configuration |
|---|---|
| CPU | Intel®Xeon®Platinum 8160CPU @2.10 GHz |
| RAM | 80GB |
| Disk | 20TB |
| Digital meter | 2500W 10A |
| OS | CentOS 7.0 |
| Spark | Cloudera spark 2.4.0-cdh6.2.1 |
| Cloudera manager | CDH 6.2.1 |
| Yarn | Yarn-3.0.0 |
| JDK | JDK 8.0 |
| Executor cores | 24 |
| Executor memory | 5GB |
| WorkNode number | 3 |

**Table 3  Benchmark workloads.**

| Workloads | Data size |
|---|---|
| PageRank | 30GB, 40GB, 50GB, 60GB, 70GB, 80GB |
| K-means | 10GB, 20GB, 30GB, 40GB, 50GB, 60GB |
| Apriori | 30GB, 40GB, 50GB, 60GB, 70GB, 80GB |

PageRank is an algorithm that determines webpage importance based on interlinking, assigning weights that depend on the number and quality of links from other pages. It is a core part of the Google search engine, helping rank pages in search results (*Gabdullin et al., 2024*; *Priyadarshini & Rodda, 2022*). K-means is a clustering algorithm that partitions data into K clusters by optimizing centroids to minimize the sum of squared distances between data points and centroids, widely used in market segmentation, image compression, and SEO (*Ashkouti et al., 2022*; *Ikotun et al., 2023*). Apriori is an association rule learning algorithm that identifies frequent itemsets and interesting associations in large datasets, used in market basket analysis, retail, e-commerce, and recommendation systems (*Dhinakaran & Prathap, 2022*; *Yan et al., 2022*).

We conducted experiments on PageRank, K-means, and Apriori algorithms with datasets of varying sizes. For PageRank, we used data sets ranging from 30GB to 80GB, with 10GB intervals (Fig. 6A). The choice of intervals allowed us to assess the efficiency of different strategies for Spark tasks and the impact of garbage collection. At 30GB, the strategy of storing data solely on the disk had the longest execution time due to sufficient memory resources, making memory a better choice for persistent storage. Our AMR strategy maintained similar execution efficiency compared to other strategies. As the data sets increased to 40GB, 50GB, and 60GB, the execution time for strategies involving only memory or memory_and_disk constantly increased at a higher rate compared to AMR, DMAOM, DMATS, and even disks. This indicated the emergence of a memory bottleneck. While the AMR strategy had slightly higher execution time, the focus of AMR

optimization was not this scale of data volume. For 70GB and 80GB data sets (Fig. 6), AMR was 41.4% more efficient than the only_memory strategy. The execution time of DMAOM changed from being lower than the only_disk strategy to higher than the only_disk strategy from 70GB to 80GB, while AMR remained 22.7% more efficient than DMAOM. AMR outperformed other strategies for large data volume pressure by effectively using memory for job acceleration without relying solely on memory.

For K-means, we used datasets from 10GB to 60GB with 10GB intervals (Fig. 6B). At 10GB, 20GB, and 30GB, the execution time for only_disk was much higher than other strategies. Although the execution time for memory_and_disk and only_memory surpassed AMR at this point, it still did not exceed only_disk, indicating the higher utility of memory acceleration over the disk. At 40GB, the execution time for only_memory and only_disk strategies was similar, while DMAOM and DMATS showed significant optimization effects. However, for data sets larger than 50GB, the execution time of the AMR strategy surpassed other strategies, demonstrating its stable execution efficiency for the pressure of large data volumes. At 60GB, the execution time of DMAOM and DMATS even exceeded that of only_disk. AMR had the shortest execution time, with notable improvements compared to other strategies.

For Apriori, datasets of the same order of magnitude as PageRank were used (Fig. 6C). At 30GB and 40GB, the execution time for memory_and_disk was lower than that of only_memory. At 50GB and 60GB, the advantage of the memory_and_disk strategy gradually decreased, and the execution efficiency of AMR was similar to DMAOM and DMATS. With 70GB and 80GB data sets, the execution time of DMAOM and DMATS became similar to only_disk or longer, while AMR exhibited significantly lower execution time compared to other strategies, improving efficiency by up to 30.1% compared to memory_and_disk. These results indicate that as the dataset scales, memory contention increases and cache acceleration decreases. The disk persistence method effectively reduces execution time by avoiding memory contention. AMR performs well in situations with large datasets resulting in insufficient memory and inefficiency.

## Garbage collection percentage & persistence hit rate

To assess the impact of garbage collection on various algorithms and datasets, we performed comparative experiments and tracked garbage collection percentages using the Spark Web UI. The garbage collection percentage represents the duration of garbage collection as a percentage of task runtime. The execution results of PageRank, K-means, and Apriori jobs are depicted in Figs. 7A, 7B, 7C.

PageRank, being computation-intensive with iterative calculations, tends to have high garbage collection percentages when using the memory_and_disk strategy by default. At 30GB, the garbage collection ratios are similar, except for a slightly higher ratio for the memory_and_disk strategy, which is still within the memory capacity limit. From 40GB onwards, the garbage collection percentages for only_memory, memory_and_disk, only_disk, DMAOM, and DMATS increase proportionally, with a slowdown at 70GB. The AMR algorithm shows minimal growth in garbage collection percentage, except for a slight increase from 30GB to 40GB. At 80GB, the garbage collection percentage of AMR

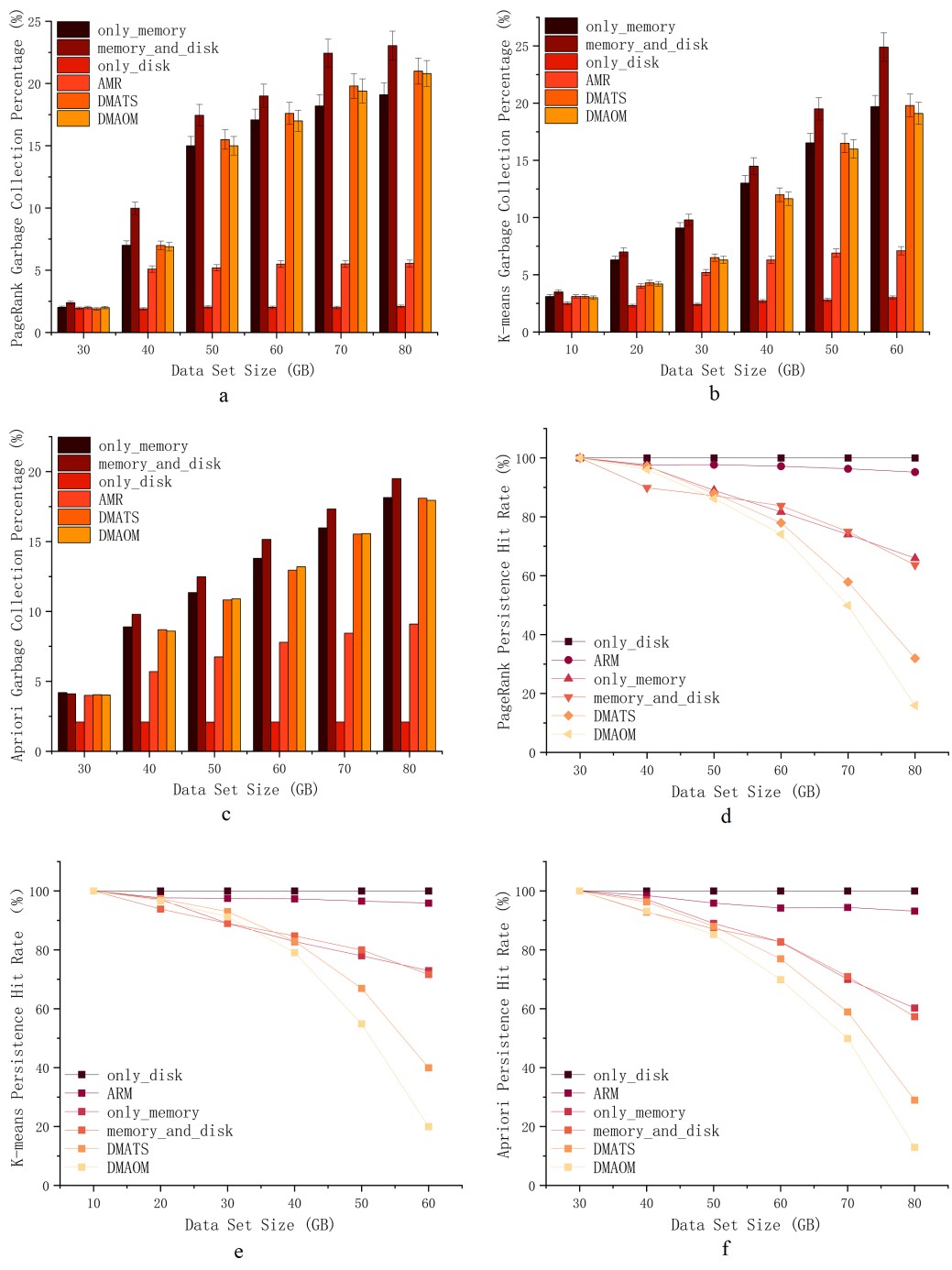

**Figure 7** Garbage collection percentage & Persistence hit rate.

is 13.6% lower than memory_and_disk due to the effective avoidance of cache contention and garbage collection as the data size grows.

For K-means, a clustering algorithm with an iterative process, the garbage collection percentage also increases with increasing data size (Fig. 7A). Overall, except for only_disk,

the AMR algorithm has consistently shown the lowest garbage collection percentage after 20GB. While only_disk has the lowest garbage collection percentage, its computation time is too long, and the advantage of garbage collection percentage alone cannot conclude its applicability to real-world scenarios. In summary, the AMR algorithm exhibits clear advantages for K-means starting from a data volume of 20GB, benefiting from the efficient utilization of memory compared to other algorithms.

Similar to PageRank and K-means, the Apriori algorithm shows similar garbage collection patterns (Fig. 7B). From 40GB onwards, as memory pressure increases, the garbage collection percentage of AMR gradually widens the gap with other algorithms. For small datasets, all strategies, including only_memory, memory_and_disk, DMAOM, DMATS, and AMR, have low garbage collection percentages. However, as the dataset size increases, these strategies experience higher garbage collection percentages and reduced job execution efficiency. Both AMR and only_disk strategies are less affected by dataset size and have lower garbage collection percentages. We can conclude that the only_memory strategy triggers more garbage collection when facing memory bottlenecks, while the memory_and_disk strategy tends to fill up memory, leading to frequent garbage collection and associated costs. In contrast, AMR flexibly caches RDDs in memory or on disk, maintaining a low garbage collection percentage.

In Fig. 7D, Eq. (10) demonstrates the significance of RDD cache hit rates. The impact of data size on persistence hit rates was analyzed by comparing DAOMA, SACM, and Spark's built-in only_memory, memory_and_disk, and only_disk strategies. The results for the PageRank job are shown in the figure. As the dataset size increases, the persistence hit rates of DAOMA, DMATS, only_memory, and memory_and_disk strategies significantly decrease. AMR after optimization, closely approaches only_disk. To further validate this, similar experimental results were obtained for the K-means execution in Fig. 7E and the Apriori execution in Fig. 7F. The analysis reveals that frequent memory replacement operations and severe contention lead to a rapid decline in hit rates. Additionally, the only_memory and memory_and_disk strategies often trigger garbage collection, which may result in cached data being cleared. AMR employs disk persistence to reduce cache replacements and garbage collection, achieving the highest hit rates.

## Energy consumption

With the increasing computing performance, server energy consumption has become a critical issue that affects computing services and enterprise revenue. To investigate energy consumption for different strategies for various algorithms, we use Power Monitor DL333502 to monitor the energy consumption of the server cluster, measured the energy consumption (kwh) of cluster during the total execution time of each task. The detailed comparative experiment was carried out as shown (Figs. 8A, 8B, 8C).

At 30GB and 40GB, only_disk exhibits higher energy consumption, while other strategies show similar energy consumption (Fig. 8A). From 50GB onwards, Spark's default strategies have relatively high energy consumption, whereas the improved strategies (AMR, DMAOM, and DMATS) show lower energy consumption. However, AMR's energy consumption advantage becomes prominent only at a data size of 60GB. At 70GB and 80GB, energy

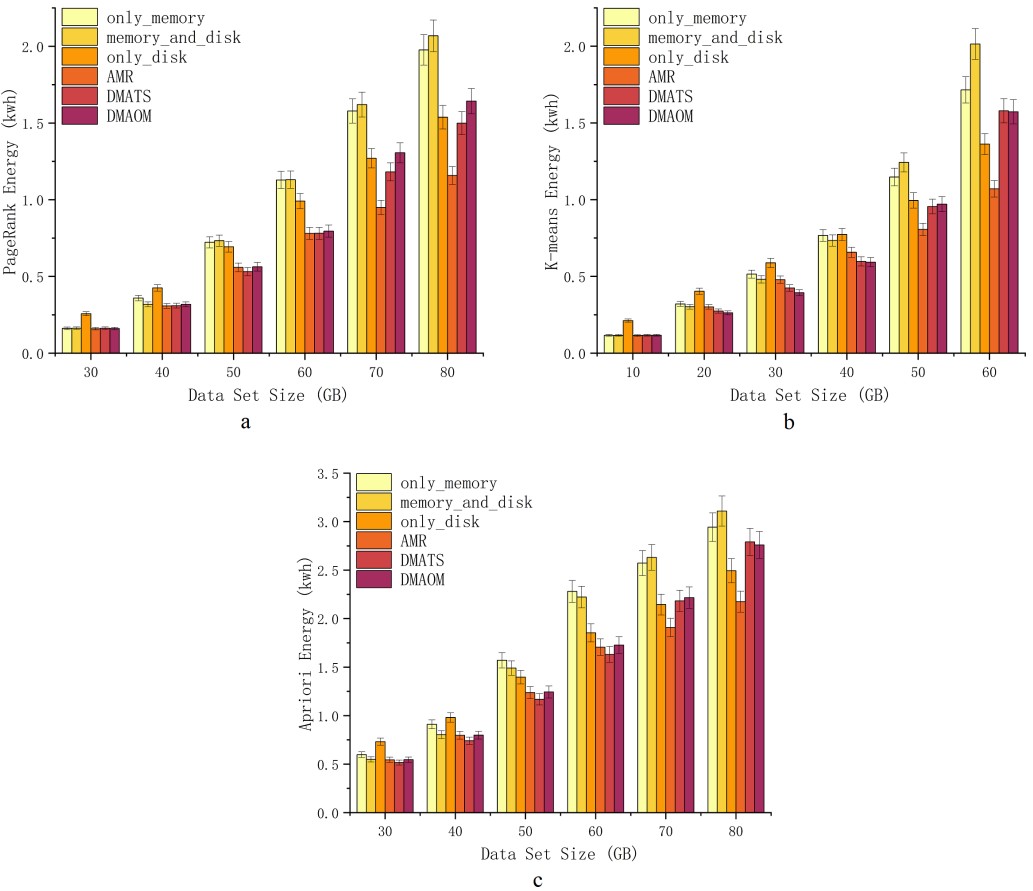

**Figure 8  Energy consumption.**

consumption rapidly increases for other strategies with the growing data volume, while the AMR algorithm demonstrates clear advantages with lower energy consumption compared to other algorithms. Notably, at 80GB, AMR's energy consumption is 44.0% lower than that of the memory_and_disk strategy, highlighting its energy efficiency advantage for large data volumes.

For the K-means algorithm, only_disk exhibits higher energy consumption at 10GB for low memory pressure, while other strategies show similar consumption (Fig. 8B). At 20GB, 30GB, and 40GB, different strategies have their advantages and disadvantages, but the improved strategies generally outperform Spark's default strategies. At 50GB and 60GB, AMR consumes less energy than all other strategies, with savings of up to 46.8% at 60GB. However, as the data volume increases, the energy consumption of DMAOM and DMATS strategies surpasses Spark's default only_disk strategy.

Similar to PageRank, the strategies for the Apriori algorithm show comparable energy consumption trends. AMR performs well with a large amount of data, demonstrating excellent data carrying capacity (Fig. 8C). Energy consumption from 30GB to 60GB is comparable to other improved algorithms. Starting from 70GB, AMR effectively utilizes

disk space, reducing memory pressure and improving execution efficiency, resulting in lower energy consumption. At 80GB, AMR's energy consumption surpasses that of DMAOM and DMATS by 22.1% and 21.2%. Notably, the energy consumption of AMR does not increase rapidly with data volume growth.

## CONCLUSION

Spark has emerged as a promising big data platform to meet the demands of large-scale data computing. Nevertheless, Spark's memory management encounters computational efficiency challenges, especially for heavy workloads scenarios. This paper introduces an adaptive memory reservation (AMR) strategy designed for heavy workloads in the Spark environment. Specifically, we model optimal task parallelism by minimizing the difference between the number of tasks completed without blocking and the actual number of tasks completed in normal rounds. Task parallelism optimal memory is determined to define an optimal execution memory space for computational parallelism. Furthermore, task parallelism is dynamically ensured in the process of Spark parallel computing through adaptive reservation of execution memory space. The allocation is adjusted by choosing to compress or increase the execution memory space based on the running tasks. Different types of RDDs are assigned appropriate storage locations to alleviate execution memory pressure, considering the cost of RDD cache location and real-time memory space occupation. Comprehensive experiments are carried out to assess the proposed AMR strategy. The results demonstrate that AMR can reduce the execution time by approximately 46.8% compared to existing memory management solutions.

Our experiments are designed to validate the experimental effects in the case of a single real cluster with multiple servers. In order to accommodate the boosting effect of multiple server clusters, the method of finding the initial adaptive task concurrency needs to be changed, which we plan to do in future work so that AMR can be applied to any Spark cluster. In addition, while better results have been achieved with high workloads, further improvements in efficiency under low loads can also be routinely performed in future work to further improve the results.

## ACKNOWLEDGEMENTS

The authors would like to thank the anonymous reviewers for their insightful comments and suggestions on improving this article.

### Funding

This work was supported by Major Science and Technology Project of Henan Province (Grant No. 201300210400), Henan Province Science and Technology Research Project (Grant No. 232102210031), Key Scientific Research Project of Colleges and Universities in Henan Province, China (Grant No. 21A520003), Key R&D and Promotion Special Project of Henan Province, China (Grant No. 212102210094 and 212102210090), and Key

Research and Promotion Projects of Henan Province (Grant No. 222102210034). There was no additional external funding received for this study. The funders had no role in study design, data collection and analysis, decision to publish, or preparation of the manuscript.

### Grant Disclosures

The following grant information was disclosed by the authors:

Major Science and Technology Project of Henan Province: 201300210400.

Henan Province Science and Technology Research Project: 232102210031.

Key Scientific Research Project of Colleges and Universities in Henan Province, China: No. 21A520003.

Key R&D and Promotion Special Project of Henan Province, China: 212102210094, 212102210090.

Key Research and Promotion Projects of Henan Province: 222102210034.

### Competing Interests

The authors declare there are no competing interests.

### Author Contributions

- Bohan Li conceived and designed the experiments, performed the experiments, analyzed the data, performed the computation work, prepared figures and/or tables, and approved the final draft.
- Xin He conceived and designed the experiments, performed the experiments, analyzed the data, performed the computation work, prepared figures and/or tables, authored or reviewed drafts of the article, and approved the final draft.
- Junyang Yu conceived and designed the experiments, performed the experiments, analyzed the data, performed the computation work, prepared figures and/or tables, authored or reviewed drafts of the article, and approved the final draft.
- Guanghui Wang conceived and designed the experiments, performed the experiments, analyzed the data, performed the computation work, prepared figures and/or tables, authored or reviewed drafts of the article, and approved the final draft.
- Yixin Song conceived and designed the experiments, performed the experiments, analyzed the data, performed the computation work, prepared figures and/or tables, and approved the final draft.
- Shunjie Pan conceived and designed the experiments, analyzed the data, performed the computation work, prepared figures and/or tables, and approved the final draft.
- Hangyu Gu performed the experiments, analyzed the data, performed the computation work, prepared figures and/or tables, and approved the final draft.

### Data Availability

The code and raw data are available in the Supplemental Files.

### Supplemental Information

Supplemental information for this article can be found online at http://dx.doi.org/10.7717/peerj-cs.2460#supplemental-information.

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
