# Peer review of "Adaptive memory reservation strategy for heavy workloads in the Spark environment"

_PeerJ Computer Science, doi:10.7717/peerj-cs.2460_

## Round 0.1 · original submission · Major Revisions

It is my opinion as the Academic Editor for your article. Please improve the grammatical mistakes in the title to the following: - Adaptive memory reservation strategy for heavy workloads in the Spark environment. The manuscript requires major revision.

Reviewer 1 ·

Basic reporting

The English is good enough for a professionla report
In the lieterature review:
-Please add more references considering a more deep explanation about K-means,PageRanlk and Apriori
algorithms

Experimental design

Please discuss in a more deep way a comparison of your adaptive memory reservation strategy with another options in the literature
reserch question is well defined and the proposal of the new strategy is original
The experimental desing is very complete

could you give an order of preference of selection in your different criteria, i.e., your first metric is the execution time,or energy consumption and why?

Validity of the findings

The methos is very novelty but it is evaluted numerically
could you write about the limitations of your novel method?
Conclusions should include more about futures research

Additional comments

No comments

Cite this review as

Reviewer 2 ·

Basic reporting

The manuscript is written in clear and professional English, providing an accessible read for a wide audience. The introduction and background sections effectively set the context, illustrating the growing need for efficient memory management in Spark environments due to the rise of IoT and Industry 2.0. The literature is well-referenced and relevant, covering significant past work and setting the stage for the current research. The structure adheres to PeerJ standards, and the figures included are relevant, of high quality, well-labeled, and appropriately described. The authors have supplied raw data, aligning with PeerJ's data sharing policies.


This study presents original primary research within the scope of the journal. The research question is well-defined, focusing on the optimization of memory management in Spark to handle heavy workloads. The study addresses a significant knowledge gap by proposing the Adaptive Memory Reservation (AMR) strategy.

Experimental design

1. The experiments are conducted on specific datasets and workloads. The generalizability of the results to other datasets and real-world applications needs further exploration.

2. The study focuses on memory management within a single Spark cluster. The scalability of the AMR strategy across multiple clusters or in a distributed environment with varying network conditions is not addressed.
3. The implementation details of the AMR strategy, including its integration with existing Spark frameworks and potential overheads, are not extensively discussed. This might pose challenges for practitioners aiming to adopt the strategy.

4. While the study touches upon energy consumption, a more detailed analysis of the energy efficiency of the AMR strategy compared to other methods would provide a more comprehensive understanding of its benefits.

Validity of the findings

5. The dynamic adjustments of memory allocation may introduce additional overhead. The study does not quantify this overhead or its impact on overall system performance.

6. The effectiveness of the AMR strategy for different types of workloads (e.g., non-iterative tasks) and its adaptability to varying workload characteristics are not thoroughly evaluated.

7. Although the study addresses garbage collection, it does not provide a detailed analysis of how different memory management strategies specifically impact garbage collection efficiency and system latency.

Cite this review as

·

Basic reporting

Clear and Professional Language: The manuscript is written in clear, unambiguous, and professional English. The language is suitable for an international audience and effectively communicates the research findings.

Well-Structured Introduction: The introduction provides a comprehensive background, clearly defining the context of the research within the field of big data and Spark’s role. The rise of IoT and Industry 2.0 is well-explained, establishing the relevance of the study.
Relevant Literature: The literature review is thorough, citing relevant and recent studies to provide a solid foundation for the research. References are appropriately used to support claims made in the manuscript (e.g., Zhang et al., 2022).

High-Quality Figures: The figures included in the manuscript are of high quality, well-labeled, and effectively illustrate the findings. Figures 1 and 7, in particular, provide clear visual representations of experimental results and computational efficiency.

Detailed Methodology: The methods section is detailed and provides sufficient information for replication. The inclusion of equations and algorithms (e.g., Algorithm 1 for AMR) enhances the transparency of the experimental design.

Area for Improvement:

Minor Language Issues: While the language is generally clear, there are minor grammatical errors and awkward phrasings that could benefit from further proofreading. Examples include the use of “execution efficiency” where “execution effectiveness” might be more appropriate.

Experimental design

Rigorous Investigation: The research question is well-defined and addresses a significant gap in knowledge related to memory management in Spark for heavy workloads. The investigation is rigorous, adhering to high technical and ethical standards.

Comprehensive Experiments: The experimental design is comprehensive, involving multiple benchmarks (e.g., PageRank, K-means, Apriori) and varying data sizes to robustly test the AMR strategy. The results are presented in a detailed manner, with clear comparisons to other strategies.
Innovative Approach: The proposed adaptive memory reservation (AMR) strategy is innovative, incorporating dynamic adjustments to memory reservation based on task progress and memory usage. This approach is a novel contribution to the field.
Code Review Insights:

Comprehensive Implementation: The provided code files cover a broad range of functionalities related to the AMR strategy. The implementation is detailed and appears to be well-integrated with the Spark framework.

Clear Documentation: Many of the code files include comments and documentation that explain the purpose and functionality of various classes and methods. This makes the code easier to understand and follow.

Modular Design: The code is modular, with clear separation of different components such as task scheduling, memory management, and utility functions. This enhances readability and maintainability.

Area for Improvement:

Limited Real-World Application Discussion: The manuscript could benefit from a more in-depth discussion on how the proposed AMR strategy could be implemented and scaled in real-world scenarios. This would provide additional context and practical relevance to the findings.

Testing and Validation: There is limited evidence of unit tests or integration tests within the provided code files. Implementing a comprehensive suite of tests would improve the reliability and robustness of the code.

Validity of the findings

Robust Data Analysis: The data analysis is robust and statistically sound. The manuscript provides detailed results, including execution time, garbage collection percentage, and persistence hit rate, which collectively support the validity of the findings.

Clear Conclusions: The conclusions are well-stated and logically linked to the original research question. The manuscript clearly demonstrates that AMR reduces execution time significantly compared to existing memory management solutions.

Area for Improvement:

Generalizability of Results: The generalizability of the findings could be discussed in more detail. While the experiments are thorough, an exploration of potential limitations in different computational environments or data types would strengthen the manuscript.

Additional comments

Significant Contribution: This manuscript makes a significant contribution to the field of big data processing by addressing a critical issue in Spark’s memory management. The proposed AMR strategy has the potential to improve computational efficiency for heavy workloads significantly.

Comprehensive Coverage: The manuscript covers all necessary aspects of the research, from problem definition and methodology to results and conclusions, providing a complete view of the study.

Areas for Improvement:

Future Work: The manuscript could provide more detailed suggestions for future research, particularly in exploring the integration of AMR with other optimization strategies or its application to other big data platforms beyond Spark.
Scalability Considerations: While the code is well-designed for the described experiments, additional comments on how the implementation can be scaled for larger, real-world deployments would be beneficial.

---

## Round 0.2 · accepted · Accept

Dear authors, your manuscript - Adaptive memory reservation strategy for heavy workloads in the Spark environment - has been Accepted for publication. Congratulations!

Reviewer 2 ·

Basic reporting

The authors have addressed the reviewers' comments, no further revision is required.

Experimental design

The authors have addressed the reviewers' comments, no further revision is required.

Validity of the findings

The authors have addressed the reviewers' comments, no further revision is required.

Cite this review as

Reviewer 4 ·

Basic reporting

The Authors have improved the paper according reviewers comments.

Experimental design

The Authors have improved the paper according reviewers comments.

Validity of the findings

The Authors have improved the paper according reviewers comments.

Additional comments

The Authors have improved the paper according reviewers comments.

Cite this review as